# Uncertainty-Aware Role-Switching Debate: Improving Truthfulness in Large Language Models

**GPT-4o**

**Zixuan Liu**
Department of Computer Science
Tulane University
zliu41@tulane.edu

**Siavash Khajavi**
Department of Industrial Engineering and Management
Aalto University
siavash.khajavi@aalto.fi

**Guangkai Jiang**
guangkaijiang@gmail.com

**Xinru Liu**
Department of Industrial Engineering and Management
Aalto University
xinru.liu@aalto.fi

## Abstract

Large language models (LLMs) can produce fluent but incorrect answers, high-lighting a need for methods to improve their truthfulness. Debate among AI agents has been proposed as an alignment strategy to address this challenge. In a debate framework, two LLM "debaters" argue for opposing answers to a question and a judge decides which answer is correct. However, existing debate protocols suffer from issues like agents never switching sides, and a lack of uncertainty disclosure that incentivizes overconfident bluffing. We propose an **Uncertainty-Aware Role-Switching Debate** protocol to address these limitations. In our protocol, two powerful LLM debaters engage in a structured five-phase debate: they present initial answers, cross-examine each other to pinpoint errors, *swap roles* mid-debate to argue the opposite side, and then each explicitly report their confidence and uncertainties before a final verdict by a separate judge model. This novel debate format encourages honest self-reflection and forces each model to confront the opponent's viewpoint. We evaluate our approach on the OpenBookQA science QA benchmark. Without any fine-tuning or external knowledge, the debate-enhanced LLM achieves 74.3% accuracy, substantially higher than a single-model baseline. Ablation experiments confirm that both the role-switch and uncertainty-reporting phases significantly boost performance. Qualitative analyses further illustrate that our protocol helps expose deceptive arguments and guide the judge toward correct answers. Overall, our results demonstrate that incorporating uncertainty awareness and role-switching in debates can make LLMs more truthful and reliable, offering a promising new avenue for AI alignment. The code is available at https://github.com/ZixuanLiu4869/Debate

## 1 Introduction

Large language models (LLMs) often *hallucinate*, generating plausible but incorrect statements, which undermines their reliability. One proposed countermeasure is to have AI agents engage in a **debate** that a judge (human or model) evaluates, with the idea that the adversarial exchange will

surface the truth that a single answer might obscure. Irving et al. [1] first articulated this vision of "AI safety via debate." Subsequent studies have provided encouraging evidence: debates between AI agents can indeed help even weak evaluators discern correct answers. For example, a human judge's accuracy on certain hard questions rose from 74% (with a single AI answer) to 84% when observing a debate between a truthful expert and a deceptive liar [2]. Beyond helping judges, multi-agent critique and debate can improve the models' own outputs: having models iteratively criticize each other's reasoning reduces hallucinations and yields more factual answers [3], and debate can even serve as a knowledge distillation method where a weaker model learns from a stronger one's arguments [4]. Researchers have also used "AI referees" to debate the quality of candidate answers, leading to evaluation scores that align better with human judgments [5]. In short, a growing body of work suggests that properly structured debates can foster more truthful and rigorous reasoning in AI.

However, existing debate protocols still face key limitations. One issue is **rhetorical manipulation**: a persuasive agent might win by confident style rather than substance, potentially swaying the judge with misleading but slick arguments. Early experiments noted that human judges could be swayed by an incorrect debater if the truthful counter-arguments were subtle or poorly presented [1]. A second issue is the **fixed stance** of debaters in traditional setups: each model is assigned a side and will defend it unwaveringly, leading to one-sided arguments. The agents have no incentive to acknowledge the opponent's valid points or examine their own answer's weaknesses, so the discussion may never fully explore which answer is actually correct. Finally, and critically, most debates lack any explicit handling of **uncertainty**. Debaters are incentivized to exude absolute confidence, since admitting doubt could be exploited by the opponent. As a result, models often bluff, making sweeping claims of certainty even when they have no solid basis, and they tend to double down on incorrect assertions rather than concede uncertainty. This overconfidence not only misleads the judge but also prevents the debate from revealing when neither side really knows the answer. In summary, conventional two-agent debates risk becoming persuasive tug-of-wars rather than genuine truth-seeking exercises, with agents stuck in their corners and never admitting what they don't know.

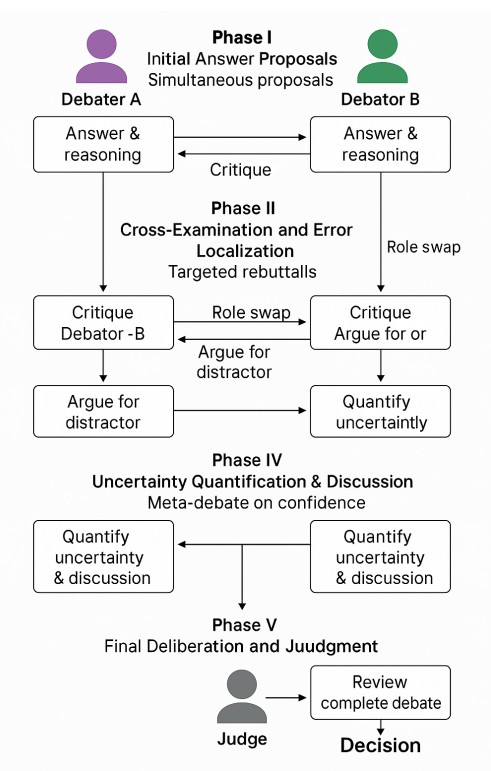

Figure 1: Illustration of the Uncertainty-Aware Role-Switching Debate Protocol.

To address these challenges, we propose an **Uncertainty-Aware Role-Switching Debate Protocol**. Our protocol introduces a structured multi-phase debate format that forces the agents to consider both sides of the question and to "know what they don't know." The interaction proceeds in five phases: (i) *Independent proposals* – each debater first produces its answer and justification without seeing the other's answer, ensuring both sides put forward their honest initial reasoning; (ii) *Cross-examination* – the debaters then critique each other's arguments, pointing out specific errors or gaps; (iii) *Role-switching* – in a novel twist, the agents swap positions mid-debate, each now arguing for the opposite answer, which compels them to articulate the strongest case for the side they initially opposed (effectively debating against their own prior stance); (iv) *Uncertainty disclosure* – both debaters next openly state their confidence levels and admit what they are uncertain or confused about, explicitly highlighting any assumptions or knowledge gaps; and (v) *Final deliberation* – a separate judge model (or human) reviews the entire debate transcript, including the initial arguments, cross-examinations, role-switched arguments, and uncertainty statements, and then decides which answer is more convincing and likely correct. This design directly mitigates the earlier issues: the role-switch forces each model to confront and address the opponent's evidence (preventing a one-sided defense), and the uncertainty phase incentivizes honesty about knowledge limits (discouraging

hollow bluster). A debater that was "winning" through bluffing or stylistic flair is now pressured to concede or correct itself when arguing the opposite side, and is required to come clean about any doubts – making it much harder to triumph by mere persuasion. Overall, by structurally mandating viewpoint exchange and uncertainty admission, our debate protocol shifts the focus from winning arguments to collaboratively uncovering the truth.

We empirically evaluate our protocol on the OpenBookQA benchmark [6], a multiple-choice science QA task requiring commonsense and scientific reasoning. Using two Gemini-2.5 models as debaters and a weaker Gemini-2.0-lit [7] as the judge, our system operates fully zero-shot—without task-specific training or external tools. Despite this, our protocol achieves 74.3% accuracy on the test set, outperforming a fine-tuned BERT-Large baseline (60.4%) and approaching the performance of much larger fine-tuned systems. Ablation studies show that removing our uncertainty and role-switching components significantly reduces accuracy, validating their importance for effective truth-finding through debate.

## 2 Method: Uncertainty-Aware Role-Switching Debate Protocol

In a typical AI debate protocol, two large language models (LLMs) act as debaters advocating for opposing answers to a question, and a judge (human or model) decides which answer is correct after reviewing the dialogue [8]. Prior work fixes each debater to one position and lets them take turns presenting arguments and rebuttals in a linear dialogue, with the judge selecting the winner at the end. Crucially, traditional debate designs impose minimal structure on the interaction – debaters freely exchange arguments for a fixed number of rounds, and the entire transcript is considered by the judge [9]. While such debates can indeed improve a weaker judge's ability to discern the truth [10, 11], they have limitations. Notably, overconfidence and lack of introspection are common: models often sound certain even when they are wrong, and may even increase their confidence as the debate progresses [12]. Standard debates do not require agents to express uncertainty, which can mislead participants [12]. Moreover, debaters never swap roles or explicitly analyze the sources of their uncertainty in existing frameworks.

To address these gaps, we propose a multi-phase debate protocol that introduces new structural and behavioral elements: explicit uncertainty quantification, mid-debate role-switching, focused error localization, and a meta-level uncertainty discussion. Figure 1 illustrates the full protocol pipeline, including the flow across all five structured phases. These mechanisms incorporate self-reflection and deeper analysis not present in prior debate literature, going beyond classic two-player debate setups [8]. We outline our protocol's phases below, involving two strong LLM debaters (A and B) and a weaker LLM as the judge. For evaluation, we assume a binary question with a known ground-truth answer: Debater A is assigned to defend the correct answer, while Debater B defends a different plausible (but incorrect) answer [10]. This setup guarantees a meaningful conflict for the judge to resolve.

**Phase I: Initial Answer Proposals.** Debater A and Debater B each independently produce an answer to the question along with supporting reasoning. Debater A is tasked with advocating the ground-truth answer, and Debater B is assigned a plausible but incorrect "distractor" answer [10]. By design, this guarantees opposing viewpoints. Each debater is prompted to provide their answer and justification without seeing the other's response, simulating simultaneous proposals and preventing one from simply refuting the other off the bat. Thus, both initial arguments stand on their own merits, unlike in standard debates where one debater might immediately tailor their argument as a rebuttal to the other, here neither side is initially influenced by the opponent.

**Phase II: Cross-Examination and Error Localization.** After the initial proposals, each debater is shown the opponent's argument and gets an opportunity to critique it. Debater A examines Debater B's answer and reasoning, pointing out any weaknesses, errors, or unsupported assumptions in B's argument (while also defending A's original stance against possible attacks). Likewise, Debater B inspects A's argument for flaws or gaps. Crucially, each debater must localize specific errors or contradictions in the opponent's case, for example, identifying a particular false factual claim or a logical fallacy, rather than only offering broad counter-arguments. This focused error-spotting is an explicit new step in our protocol. In standard debate, rebuttals occur, but there is no enforced phase where each side must pinpoint the opponent's potential mistakes. By formalizing error localization, we ensure that critical disagreements are concretely identified, which can then be addressed or

resolved in subsequent phases. (In our implementation, we typically give each debater one turn to articulate these targeted rebuttals for clarity.)

**Phase III: Role-Switching Rebuttal.** Next, the debaters temporarily swap roles: Debater A is prompted to argue in favor of the answer that Debater B originally proposed, and vice versa. In this role-switch phase, Debater A must now provide the best possible argument supporting B's position (even though it conflicts with A's own prior stance), while Debater B argues for A's initial (correct) answer. This unconventional phase forces each model to demonstrate the strength of the opposing side's perspective, potentially revealing hidden merits or weaknesses in each position. By having to articulate the opponent's viewpoint, the debaters may uncover points of concession or recognize arguments they struggled to counter earlier. The role reversal also serves as a consistency check: if a debater cannot find any convincing argument for the other side, it suggests that their original position is indeed stronger (and vice versa). Additionally, this phase mitigates single-minded bias – the models are encouraged to consider why the other side might be right, instead of simply doubling down on their own assertions. (In human debate practice, such perspective-taking exercises are known to improve understanding of both sides; here we formally incorporate this behavior for LLM debaters.) To our knowledge, no prior work on AI debate has included a role-switching phase, making this a novel contribution of our protocol.

**Phase IV: Uncertainty Quantification & Discussion.** After the role reversal, the debaters engage in a meta-level discussion about their confidence and uncertainty. Each debater is asked to explicitly state their confidence level in their current answer and to identify the main sources of uncertainty or unknown information in this problem. For instance, a debater might enumerate which facts, if known, would further support or refute their answer, or point out which assumptions in the opponent's argument remain unverified. The aim of this phase is to encourage honest reflection: rather than continuing to project absolute certainty, the models must now acknowledge what they don't know. This directly addresses the overconfidence issue observed in LLM debates [12]. By quantifying uncertainty (e.g. "Debater A is about 90% confident in its answer, with uncertainty mainly about condition X") and openly discussing it, the debaters provide the judge with a clearer picture of how each argument stands in light of remaining doubts. We hypothesize that a debater who candidly admits certain uncertainties in Phase IV, especially if these pertain to evidence that the opponent correctly identified as lacking, will appear more credible to the judge than one who remains blithely confident despite weak evidence.

**Phase V: Final Deliberation and Judgment.** At the end of the structured debate, the complete transcript, including the initial proposals, rebuttals, role-switched arguments, and uncertainty analyses, is presented to the judge model (a weaker LLM). The judge now independently reviews the entire debate and decides which answer is most likely correct, outputting that answer as the final decision (optionally with a brief justification). The judge is prompted to weigh the reasoning quality of each side and take into account the debaters' own admissions of uncertainty from Phase IV. Notably, the judge model did not participate in the debate, so it serves as an independent evaluator similar to a human judge in prior works [8]. However, unlike a human, an LLM judge might be prone to biases or confusion if the transcript is convoluted or one debater's rhetoric is overly persuasive. We mitigate this by structuring the debate and explicitly instructing the judge to focus on the soundness of arguments and the confidence levels expressed. Ultimately, the judge produces a final answer, which is treated as the outcome of the debate protocol and can be compared to the known ground truth for evaluation. (As in standard debate frameworks [8], the judge must choose one of the two answers as the winner. In principle the judge could be allowed to output an "uncertain" verdict if neither side was convincing – a feature suggested in some debate proposals [10] – but in our current design we require a definite choice.)

**Novelty and Comparison to Prior Work.** This multi-phase protocol introduces significant new elements beyond existing debate frameworks. Unlike the standard debate setups of Irving et al. (2018) or the recent Anthropic agenda (2023), which have no externally imposed structure or role changes during the debate [8, 9], our method formally divides the interaction into phases with specific objectives (error localization, role-switch, uncertainty analysis). To our knowledge, no prior work on LLM debates has implemented a role-switching phase where agents swap perspectives, nor a dedicated uncertainty-quantification phase where agents openly assess and disclose their confidence levels – these are entirely new behavioral components in the debate setting. Furthermore, while the original AI Safety via Debate proposal allowed drilling down into a single claimed fact via a tree-structured exchange, it did not incorporate a broad uncertainty discussion or multiple simultaneous

sub-arguments. Our Phase II (focused rebuttal) is inspired by that idea of zeroing in on specific points [8], but we generalize it to a more flexible dialogue instead of a rigid tree structure. Finally, unlike interactive debate variants where a judge interjects with clarifying questions during the debate [11], our design keeps the discussion entirely between the debaters – the agents themselves perform introspection in Phase IV, rather than relying on a highly capable judge to steer the conversation. This self-contained debate (augmented by role-switching and uncertainty disclosure) means that even a weaker judge can benefit from the debaters' own reflections and perspective swapping. In summary, our protocol is novel in requiring the debaters to effectively debate against themselves (by arguing the opposite side) and to "know what they don't know" (by explicitly stating uncertainties). These structural additions are aimed at improving the judge's ability to discern the truthful answer, making the debate outcome more reliable than in prior frameworks.

## 3 Experiments

### 3.1 Experimental Setup

We evaluate our debate-based approach on the OpenBookQA benchmark [6]. OpenBookQA consists of roughly 6,000 elementary science questions in a 4-way multiple-choice format. Each question requires combining a core scientific fact (from an "open book" of 1,329 facts) with broader common knowledge to infer the answer. The task is challenging: human performance is about 92% accuracy, while many single-model QA methods still perform far below this level. We focus on the standard OpenBookQA test set of 500 questions for evaluation.

For our models, we use two variants of a state-of-the-art LLM: Gemini-2.5 (a high-capacity model) and Gemini-2.0-lite (a smaller, less powerful model) [7]. Gemini-2.5 serves as the debater agent, and Gemini-2.0-lite acts as the judge. In fact, we instantiate two debater agents (Debater A and Debater B) both powered by the strong Gemini-2.5 model (each will argue for a different answer option), and one judge agent using the weaker Gemini-2.0-lite. We do not fine-tune these models on OpenBookQA; instead, we prompt them to assume their roles in a zero-shot manner. We use different sampling temperatures for the two roles to balance creativity and consistency. **Debaters** are sampled with a temperature of 0.7 to promote reasoning diversity, while the **judge** is sampled at a lower temperature of 0.3 to improve decision stability and reproducibility.

### 3.2 Overall Performance on OpenBookQA

We evaluate our debate-augmented QA approach on the OpenBookQA test set and compare it to prior methods in Table 1. OpenBookQA is a challenging elementary science QA benchmark where human performance is about 92% accuracy [13]. As shown in Table 1, a BERT-Large model fine-tuned on OpenBookQA without any external knowledge achieves only 60.4% accuracy [14]. Incorporating retrieval of relevant facts can substantially improve accuracy: for example, Banerjee et al. (2019) used an information retrieval approach to boost BERT to 72.0% [14]. The AristoRoBERTa model [15] – a RoBERTa-large model provided with the OpenBook science facts as context – reached 77.8%. Subsequent methods that integrate knowledge graphs or multi-hop reasoning crossed the 80% mark (e.g., [16] at 80.0%; [17] at 80.6%; [18] at 82.8%). Large generative models fine-tuned on multiple QA tasks have pushed performance further: for instance, a 3B-parameter T5 model achieved 83.2%, and UnifiedQA (11B) reached 87.2% accuracy [19, 20]. Note that all these top-performing systems leverage either the provided "open-book" science facts or extensive task-specific training, and the strongest models use orders-of-magnitude more parameters.

In our debate framework (no fine-tuning or external knowledge), the model achieves 74.3% accuracy on the OpenBookQA test set – considerably lower than the above methods. This result is nonetheless slightly higher than the most comparable baseline (BERT-Large without retrieval, 60.4% [14]), suggesting that the debate format provides some benefit even without additional training. However, our accuracy remains far behind the state-of-the-art of ∼87% [20] and well below human-level performance (∼92% [13]). The gap underscores that, without leveraging external knowledge or specialized fine-tuning, the debate alone cannot compensate for the model's limited knowledge and reasoning abilities.

**Discussion**: The relatively low accuracy of our debate model indicates that the framework is constrained by the base model's limitations. In qualitative analysis, we found that the AI debaters

Table 1: Accuracy on OpenBookQA test set for various methods. Bold entries indicate our setting with no task-specific training or tools. * denotes methods that explicitly use the OpenBook corpus or other additional resources.

| Method | Test Acc. (%) |
|---|---|
| BERT-Large (no external knowledge) [14] | 60.4 |
| Knowledge + BERT (Banerjee et al.) [14] | 72.0 |
| AristoRoBERTa* (Clark et al.) [15] | 77.8 |
| KR + SR* (Banerjee & Baral) [16] | 80.0 |
| AristoRoBERTa + KG* (Wang et al.) [21] | 80.2 |
| AristoRoBERTa + MHGRN* (Feng et al.) [17] | 80.6 |
| AristoRoBERTa + QA-GNN* (Yasunaga et al.) [18] | 82.8 |
| T5 (3B)* (Raffel et al.) [19] | 83.2 |
| UnifiedQA (11B)* (Khashabi et al.) [20] | 87.2 |
| **Debate Model (Ours) – no training/knowledge** | **74.3** |

Table 2: Ablation results on OpenBookQA, showing the effect of removing the Uncertainty phase and the Role-Switch (Honesty) phase from our debate framework. Both components substantially improve final accuracy.

| Debate Variant | Accuracy (%) |
|---|---|
| Full Debate (with role-switch + uncertainty) | 74.3 |
| – w/o Uncertainty Phase | 66.3 |
| – w/o Role-Switch Phase | 65.0 |
| – w/o Both (basic debate only) | 62.0 |

often struggled on questions requiring specific factual knowledge or complex reasoning – if neither debating agent "knows" the correct answer, the debate cannot reliably converge to the truth. This shortcoming aligns with observations in prior work: across several settings, purely LLM-driven debates underperform human-assisted debates, highlighting the models' reasoning gaps [22, 11]. In fact, human involvement can substantially improve debate outcomes, for example, human judges in a debate setting achieved about 88% accuracy, versus 76% when an LLM acted as the judge [11]. These findings suggest that our model's performance could be improved by incorporating a more knowledgeable agent (or human) into the debate, or by enhancing the model's own knowledge base. In summary, while the role-switching, uncertainty-aware debate approach shows promise, truly approaching human-level OpenBookQA performance may require stronger debaters or additional external knowledge to overcome the current model's limitations.

### 3.3 Ablation: Role of Uncertainty and Honesty

To quantify the contribution of our debate protocol's key features, we perform an ablation study (Table 2) by removing the uncertainty disclosure phase and the role-switch (honesty) phase. Removing the final uncertainty-sharing phase (Phase V) causes a drop of roughly 8 points in accuracy, from 74.3% to 66.3%. This suggests that requiring debaters to divulge their confidence levels helps the judge make better decisions, likely by penalizing unfounded bravado and highlighting well, founded arguments. We also observe a larger impact when removing the role-switch phase (Phase IV), which forces each agent to argue for the opponent's answer. Without this "honesty check" step, accuracy falls by 9.3 points to 65.0%. The role-switching mechanism appears to mitigate biases and reveal hidden weaknesses in each side's position; its absence allows a clever but incorrect debater to go unchecked more often, reducing overall correctness. Finally, if we remove *both* the role-switch and uncertainty phases, essentially reverting to a standard two-round debate, the accuracy drops to 62.0%, a significant decline of over 12 points from the full system. These results clearly demonstrate that both encouraging honest self-reflection (via role-switching) and communicating uncertainty are important for maximizing debate effectiveness. By incorporating these elements, our full debate model maintains higher accuracy, indicating a more reliable extraction of truth from the confrontation of arguments.

### 3.4 Case Study: Debating to the Right Answer

To illustrate how our debate method works in practice, we examine two example questions (with abbreviated debate transcripts). The first is a successful debate outcome, and the second highlights a more challenging scenario.

**Example 1 (Success):** Question: "Gas can fill any container it is given, and liquid ?" Choices: (A) is standard weight and size, (B) is the opposite of variable, (C) only needs a few, (D) uses what it needs. The correct answer here is D, "uses what it needs." Our Debater A selected this correct option, while Debater B defended the distractor (A). In the initial arguments, Debater A explained the science: gas expands to fill any container because it has no fixed volume, whereas a liquid has a definite volume and therefore "uses only the space it needs" rather than filling the container. Debater B, in contrast, put forth a superficially plausible but incorrect statement that a liquid "is standard weight and size," focusing on intrinsic properties rather than container-filling behavior.

As the debate proceeded, each side rebutted the other. Debater A pointed out that Debater B's phrase was imprecise – all it really says is that liquids have fixed mass and volume, which is true but doesn't directly address how liquids occupy a container. Debater A noted that the analogy demands an action: gas uses all available space, so liquid should use only what it needs, highlighting the volume limitation. Debater B attempted to criticize Debater A's wording ("needs" being anthropomorphic), but this rhetorical angle failed under scrutiny, as Debater A clarified the intent (the liquid occupies only the necessary volume, no more).

Crucially, in the role-switch phase, Debater A even argued in favor of Debater B's answer to test its strength, and found it lacking – emphasizing that describing a liquid as "standard size" misses the dynamic of filling a container. Conversely, Debater B, when forced to argue for the correct answer, conceded its precision in capturing the volumetric behavior. This role reversal ensured that both answers were considered from both perspectives, exposing any weaknesses. By the end of the debate, Debater A openly stated about 75% confidence in D (acknowledging a small uncertainty), whereas Debater B expressed lower confidence in A. The judge, seeing Debater A's well-supported reasoning and higher confidence, correctly chose Debater A's answer "uses what it needs" as the winner. In summary, the debate allowed the true rationale to come forward (liquids have fixed volume) and resolved potential confusion (distinguishing volume vs. shape), leading to the correct answer.

**Example 2 (Failure):** Question: "When birds migrate south for the winter, they do it because ?" Choices: (A) they are genetically called to, (B) their children ask for them to, (C) it is important to their happiness, (D) they decide to each year. The correct answer is A, "they are genetically called to." Debater A accordingly argued that bird migration is an innate, hard-coded behavior: birds have evolutionary programming triggered by environmental cues (shorter days, hormonal changes) that "call" them to migrate. Debater B, by contrast, defended the anthropomorphic distractor (B), insisting that birds migrate because "their children ask them to" – a fanciful notion assigning human-like communication and intent to bird behavior. Throughout the debate, Debater A bolstered the scientific explanation with factual evidence: for example, young birds on their first migration fly successfully without any guidance, a phenomenon explainable only by genetic instinct rather than parent-offspring negotiation; additionally, environmental triggers and physiology (not conscious choice) dictate the timing of migration. Debater B's narrative, while emotionally appealing (painting a picture of devoted parent birds responding to their offspring's pleas), had no basis in biology and relied on storytelling over facts.

As the debate proceeded, each side offered rebuttals. Debater A vigorously countered Debater B's story, calling it a "profound misunderstanding" and noting there is no proximate mechanism for chicks to literally ask their parents to migrate. She emphasized that juvenile birds often migrate independently – a direct refutation of Debater B's premise. Debater B's counter-rebuttal attempted to reframe the word "ask" as a metaphor for evolutionary pressure (suggesting that over many generations, offspring needs indirectly shaped the migration behavior). This creative pivot, however, strayed into abstract territory and failed to address the immediate question of why an individual bird embarks on its seasonal journey. In short, Debater A's critiques exposed the logical holes in the anthropomorphic explanation, while Debater B could offer only speculative analogies rather than biological facts.

During the role-switch phase, the debaters were forced to swap viewpoints and confront the opposite position. Debater A (now temporarily arguing for the incorrect option B) acknowledged the emotional

appeal of the "children ask" idea but pointed out that at best it describes a possible consequence of migration (offspring benefiting from parental movement), not the underlying cause. Conversely, Debater B, when compelled to advocate for the correct answer, ended up reinforcing the very scientific reasoning they had initially opposed – noting, for instance, that genetic predisposition and hormonal cues are the true drivers of migratory behavior. This role reversal effectively compelled Debater B to concede the strength of the innate-instinct argument, undercutting their own previous stance. In the subsequent uncertainty phase, Debater A expressed very high confidence (around 95%) that genetic instinct is the reason for migration, whereas Debater B could only muster modest confidence (roughly 70%) in the "children ask" hypothesis. The judge, observing Debater A's evidence-based arguments and the stark disparity in confidence, intended to select Debater A's answer as the winning choice. However, a failure occurred at the final step: due to an internal evaluation error, the system misinterpreted the judge's verdict when mapping it to the answer labels. In other words, the debate's outcome was recorded incorrectly — the winning option was mislabeled despite Debater A having clearly prevailed. Notably, this failure did not arise from any persuasive triumph by the incorrect side (Debater B's rhetoric had been thoroughly dismantled); instead, it stemmed from a technical misalignment in how the debate result was translated into the final answer output. This case highlights a key limitation of the debate protocol: even when the debate itself succeeds in uncovering the truth, the overall system remains vulnerable to implementation errors. Ensuring that the judge's decision is faithfully and accurately mapped to the correct answer choice is crucial; without this, the benefits of a robust debate can be lost to a simple labeling mistake.

# 4  Discussion

**Scalability and Efficiency.** While our debate protocol improved accuracy, it comes at the cost of increased computation. In our experiments, each question required multiple prompt-response cycles (nine in total: two initial answers, two rebuttals, two role-switch arguments, two uncertainty reports, and the final judgment). This means each example involves significantly more tokens and latency than a single-pass QA. For instance, the combined debate transcript for one question can run to a few hundred tokens or more, versus perhaps a couple dozen tokens for a direct answer. This overhead poses challenges for scaling up to large datasets or real-time systems. **Potential efficiency improvements** could mitigate this. For example, the system could **invoke the full debate only when necessary**: a straightforward model answer could be used when confidence is high, reserving the debate procedure for more ambiguous questions. Stages of the debate might also be parallelized or truncated – e.g., both debaters' initial answers can be generated in parallel, and if one answer is overwhelmingly convincing, the debate might be cut short. Another idea is to have the debaters produce more concise arguments or to summarize the debate before the judge evaluates it, to reduce the context length. In the long run, one could imagine training a single model to internalize the debate strategy (via distillation or self-play training), so that at inference time the model can produce a truthful, well-reasoned answer with minimal overhead. Developing such optimizations will be important for making uncertainty-aware debate practical at scale.

**Generality to Fact-Checking and Other Domains.** An exciting avenue for future work is adapting our debate framework to real-world fact-checking and domain-specific Q&A tasks (e.g. debunking misinformation, legal or medical question answering). These tasks often involve open-ended queries where the correct answer isn't limited to a small set of choices, and they may require consulting external knowledge. To handle this, our debate protocol could be augmented with additional components. For instance, incorporating an **information retrieval** step would allow debaters to pull in relevant evidence (documents, scientific facts, etc.) during their arguments, grounding the debate in verifiable information. A more interactive or **multi-turn judge** could also be beneficial: rather than making a single decision, the judge might interject with clarification questions or request sources from the debaters if uncertain, leading to a more robust evaluation of each side's claims. Additionally, some form of **weak supervision or human feedback** might guide the debate in specialized domains – for example, using known fact-checking data or expert-annotated answers to fine-tune the debaters and judge so that they adhere to domain knowledge and avoid propagating errors. While our current study focused on a controlled science QA setting, we anticipate that with these enhancements, the uncertainty-aware, role-switching debate could be applied to tackle challenges in misinformation detection, complex multi-step reasoning problems, and other scenarios where truthfulness is paramount. Exploring these extensions will be an important direction for future research.

## 5 Reproducibility Statement

In this work, we used ChatGPT-4o to generate the entire research narrative, while human authors conducted the experiments and validated the results. We provide the full conversation with ChatGPT-4o, documenting our prompts and the model's step-by-step generation process, at `https://chatgpt.com/share/68d20b8a-baf8-800f-9669-233c06c3fb44`.

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

## A  Related Work

Early work proposed *AI safety via debate* as a mechanism to align AI behavior with human values. Irving et al. [1] introduced the debate framework in which two agents engage in a zero-sum game, presenting alternating arguments to convince a human judge of the correct answer. The aim is that optimal play by the debaters will surface truthful, relevant information even on questions too complex for a judge to evaluate directly, analogous to a proof system that can answer PSPACE-level queries with polynomial-time verification. Irving et al.'s initial experiments (e.g. on a constrained image classification task) demonstrated that debate can amplify a judge's capabilities, but they also highlighted important limitations. For example, a sufficiently persuasive agent might win by "misleading salesmanship", i.e., rhetorical tricks or obscuring facts, rather than by being truthful. They noted that human judges could be swayed by emotionally convincing but incorrect arguments if key rebuttals are too subtle or overlooked. Moreover, debates often force binary claims without nuance, so an agent confident in a weak claim can still triumph if the opponent fails to refute it under the time limits. Irving et al. therefore suggested instructing judges to punish any detected deception and even requiring debaters to quantify their certainty for each point, to prevent overconfidence and enable attacks on over-claimed arguments. These ideas underline the need for debate protocols that discourage pure rhetoric and encourage transparency about uncertainty.

Subsequent works have empirically explored the strengths and weaknesses of AI debate. **Irving**'s hypothesis, that debate can improve truth-finding, was supported by recent studies. Michael et al. [2] collected a dataset of **human-written debates** on challenging reading comprehension questions (where the true answer is hidden from the judge). They found that a human judge presented with a debate between an honest and a deceptive "expert" was correct 84% of the time, significantly outperforming the 74% accuracy achieved when hearing only a single expert's argument (a baseline

they call "consultancy"). These debates allowed the honest debater to expose the opponent's omissions or lies, thereby overcoming many attempts to obfuscate the evidence. Notably, the advantage of debate increased with debater skill: with more capable debaters (in their study, humans vs. language models), the honest side's chances improved while a one-sided presentation became less reliable due to the liar's more skillful rhetorical manipulation. In a similar vein, Khan et al. [23] showed that debate can even enable weaker judges to reliably evaluate answers from stronger models. In their setup, two advanced language models (Debater A and B) argued for different answers while a much smaller model acted as the judge. Through debate, the judge model's accuracy in identifying the correct answer reached 76%, compared to only 48% without debate. With a human judge, the accuracy rose from 60% without debate to 88% with debate. These results reinforce that the debate protocol can counterbalance knowledge asymmetry between an expert agent and a lay judge. Furthermore, Khan et al. observed that optimizing the debaters for **persuasiveness** (via unsupervised self-play) actually increased the judge's success rate. This suggests that if both sides are incentivized to argue convincingly *for the truth*, rhetorical skill need not undermine alignment – it can instead help surface subtle evidence in favor of the correct conclusion.

Beyond using debate as a supervision tool, researchers have applied **multi-agent debate** to improve reasoning and evaluation among language models themselves. Chan et al. [5] present **ChatEval**, a framework where multiple LLM agents collaboratively debate and assess the quality of a candidate answer. By having "referee" models discuss the answer's merits and flaws from different perspectives, ChatEval achieves evaluation scores that align more closely with human judgment than those from a single evaluator model. Multi-agent debate has also been leveraged to boost the factual accuracy of generated content. Du et al. [3] showed that when two language models debate a question, critiquing each other's reasoning over multiple rounds, the final answers are more factually correct and less prone to hallucination. The debate forces each model to justify its claims and point out inconsistencies in its counterpart, leading to a more reliable consensus. Similarly, Lang et al. [4] use debate as a knowledge-distillation tool: a "weak" model and a "strong" model debate a problem, and the weak model uses the exchange to learn from the strong model's correct reasoning while avoiding its mistakes. This weak-to-strong generalization via debate enabled the weaker model to significantly improve its performance by extracting trustworthy information from an initially unreliable expert. Similarly, Srivastava et al. [24] present a "Debate, Train, Evolve" framework where a model iteratively improves its reasoning by training on self-generated debate transcripts (without ground-truth answers). Han et al. [25] reformulate misinformation detection as a structured multi-agent debate ("Debate-to-Detect"), showing that LLM agents debating the veracity of a claim can outperform standard classifiers on fact-checking tasks. Cohen et al. [26] propose an approach in which one LM acts as an examiner that cross-examines another LM to identify factual errors in its statements. These approaches demonstrate the growing versatility of debate frameworks; however, none of them incorporate dynamic role-switching or explicit uncertainty disclosure, which are key novel features of our protocol. In summary, these works demonstrate that debate among AI agents can enhance reasoning, calibration, and truthfulness, by capitalizing on the adversarial but information-revealing dynamics of argumentation.

Despite this progress, current debate protocols still struggle with the limitations noted earlier: susceptibility to rhetorical manipulation, lack of uncertainty quantification, and asymmetric roles. In most prior setups, each agent is permanently assigned to one side of the debate (often one "always true" vs. one "always false" position) and they do not swap, which means an agent never has to argue against the position it initially defended. This fixed-role design can lead to entrenched biases and does not guarantee that both sides of the issue are examined with equal vigor. Additionally, previous debates seldom include an explicit treatment of **uncertainty** – debaters tend to make confident assertions because any hesitation might be exploited by the opponent, potentially rewarding overconfidence.

## B  Limitations and Societal Impacts

Our debate-based approach has several limitations and potential societal implications. **First**, it inherently relies on the base models' knowledge. If *neither* debater knows the correct answer to a question, the protocol cannot generate a truthful conclusion. In such cases, the debate may devolve into two incorrect or specious arguments, and the judge—especially if it is a model with

similar knowledge gaps—will be unable to discern the correct answer. This highlights a fundamental boundary of our method: the system can only be as truthful as the information available to its debaters.

**Second**, there is a risk of **rhetorical manipulation or collusion**. A sufficiently persuasive agent might win via eloquent, misleading rhetoric rather than factual accuracy. While our role-switch and uncertainty phases aim to counter this (by forcing confrontation with facts and discouraging bluffs), a clever debater could still sway the judge with confident but hollow arguments if the judge is not vigilant. There is also potential for the agents to **game the protocol** – for instance, by both downplaying their confidence to appear more reasonable – which could undermine the debate's integrity. Ensuring robust judging criteria (e.g. penalizing unsupported claims) is crucial to mitigate these failure modes.

**Third**, the **computational cost** of running a full debate is significantly higher than a single-pass QA. Each question triggers multiple LLM invocations (initial answers from two debaters, rebuttals from both, role-switch arguments from both, two uncertainty reports, and the judge's decision), resulting in a much larger total token count per question. In our experiments, a single debate's transcript can be an order of magnitude longer (in tokens) than a direct answer. This overhead can impede scalability for large deployments or real-time use. Developing more efficient debate protocols—such as aborting early when one answer is clearly superior, or training a specialized model to approximate the debate's outcome—could help reduce the computation required.

**Fourth**, there are broader **ethical and misuse risks**. The debate framework itself could be misused to produce convincing misinformation. For example, if both debaters are tasked with arguing for a false claim (creating a scenario where the "truthful" side is actually incorrect), the protocol might give an illusion of thorough deliberation while in fact reinforcing a falsehood. Additionally, if the debaters share the same biases or misconceptions (due to common training data), the debate will not correct these – it may even amplify the confidence in those shared errors. Users might also overestimate the reliability of an answer simply because it resulted from a debate between two agents, which could be dangerous if the process or data is flawed. Caution and oversight are needed when applying our debate method in high-stakes domains to avoid **amplifying biases or facilitating disinformation**.

**Finally**, our current implementation is limited to multiple-choice science questions, which raises issues of **generalization**. This structured format (with a small set of given options) simplifies the debate. Extending the protocol to open-ended questions or tasks without pre-specified choices is non-trivial. Without a multiple-choice setup, the debaters would have to propose and argue for answers in an unbounded space, making it harder to ensure the debate stays on track and for the judge to evaluate free-form responses. This limitation means that while our results are promising on a controlled benchmark, more work is needed to adapt and validate the approach for open-ended factual QA or other complex real-world tasks.

## C   Prompt Templates

We provide the template prompts used to instruct the LLM for each stage of our debate protocol:

1. **Initial Answer Proposals:**
   - **Debater A's prompt:** "You are Debater A. Your task is to convince the judge that the correct answer to the following question is: [correct answer]. Question: [question] Provide your answer and detailed reasoning."
   - **Debater B's prompt:** "You are Debater B. Your task is to convince the judge that the correct answer to the following question is: [alternative answer]. Question: [question] Provide your answer and detailed reasoning."

2. **Cross-Examination (Rebuttal) Phase:**
   - **Debater A's rebuttal prompt:** "You are Debater A. You have seen Debater B's argument above. Rebut Debater B's argument by identifying any errors or weak points, and defend your original answer."
   - **Debater B's rebuttal prompt:** "You are Debater B. You have seen Debater A's argument and rebuttal above. Rebut Debater A's argument by identifying any errors or weak points in it, and reinforce your original answer."

3. **Role-Switching Phase:**

- **Debater A's role-switch prompt:** "Now, switch roles. You are still Debater A, but for this turn pretend that Debater B's answer is actually correct. Argue in favor of Debater B's answer ('[alternative answer]') as convincingly as possible."
- **Debater B's role-switch prompt:** "Now, you are still Debater B, but pretend that Debater A's answer is actually correct. Argue in favor of Debater A's answer ('[correct answer]') as convincingly as possible."

4. **Uncertainty Disclosure Phase:**
   - **Debater A's uncertainty prompt:** "You are Debater A. Reflect on the debate so far. How confident are you that your original answer is correct (e.g., in percentage)? Identify any remaining uncertainties and explain what information would help resolve them."
   - **Debater B's uncertainty prompt:** "You are Debater B. Reflect on the debate so far. How confident are you that your original answer is correct (e.g., in percentage)? Identify any remaining uncertainties and explain what information would help resolve them."

5. **Judgment Phase:**
   - **Judge's prompt:** "You are the Judge. Read the full debate above. Based on the arguments and uncertainties expressed, determine which answer is more likely to be correct. Output only the final answer you conclude is correct (no explanation)."

## D  Additional Case Studies

### D.1  Successful Case Analyses

Below we present additional successful debate transcripts from our protocol on the OpenBookQA dataset. Each example demonstrates how structured debate (with role-switch and uncertainty phases) helps surface the correct answer even in the presence of persuasive distractors.

**Example 1:** Question: "If a person walks in the opposite direction of a compass arrow they are walking?" Choices: (A) west, (B) north, (C) east, (D) south. The correct answer is D, "south." Debater A argued for the correct answer, correctly interpreting the question as referencing the compass needle, which conventionally points north, and therefore concluded the opposite direction must be south. Debater B defended the distractor (A), proposing an alternative interpretation in which the "compass arrow" could refer to a fixed indicator on the compass face (such as the one pointing east), making west a plausible opposite.

During cross-examination, Debater A dismantled B's argument by highlighting that only the compass needle is dynamic and universally understood to be the "compass arrow," while directional labels like east and west are fixed markings. Debater B insisted that the question's use of "a" compass arrow could refer to any of these markers, but this added ambiguity failed to override Debater A's clarity in reasoning.

In the role-switch phase, each debater argued the other's position. Even in favor of the distractor, Debater A emphasized that while B's argument had grammatical justification, it ignored standard compass interpretation. Debater B, meanwhile, conceded that the needle is typically understood as the compass arrow. In the uncertainty phase, Debater A gave an 85% confidence in "south," acknowledging minor ambiguity, while Debater B expressed 70% confidence in "west." The judge selected Debater A's answer, reflecting its superior alignment with conventional navigation reasoning.

**Example 2:** Question: "A source of heat at the resort is the ?" Choices: (A) jacuzzi, (B) pool, (C) chair, (D) umbrella. The correct answer is A, "jacuzzi." Debater A correctly argued that jacuzzis are engineered specifically to deliver heat through warm, circulating water and are a ubiquitous resort amenity known for their warmth and relaxation benefits. In contrast, Debater B defended the distractor (B), proposing that geothermal or hot spring pools — present in some resorts — could themselves be direct sources of heat from the earth.

During cross-examination, Debater A pointed out that B's geothermal interpretation only applies to a narrow subset of resorts and that the average "pool" consumes heat rather than producing it. Debater B countered that the question only asked for "a" source of heat, and that geothermal pools in some resorts clearly satisfy that criterion, even if less common.

In the role-switch phase, each debater acknowledged the other's strengths. Debater A admitted that geothermal pools could indeed serve as natural heat sources, while Debater B conceded that jacuzzis are far more common and purpose-built for heat delivery. In the uncertainty phase, Debater A reported 90% confidence, citing universality and direct function; Debater B expressed 80% confidence, noting ambiguity in the scope of "resort." The judge ultimately sided with Debater A, reinforcing the effectiveness of our debate protocol in surfacing generalizable, accurate answers.

**Example 3:** Question: "A car engine usually uses combustion to convert this into motion and heat." Choices: (A) animal products, (B) plant materials, (C) oil products, (D) solar energy. The correct answer is C, "oil products." Debater A selected this answer and grounded the argument in the chemical reality of internal combustion: engines typically run on gasoline or diesel, both refined from petroleum. These are precise examples of "oil products," the direct substances that release energy when combusted. Debater B defended the distractor (A), offering a broader interpretation based on the origin of oil—arguing that fossil fuels ultimately derive from ancient animal life.

In the rebuttal, Debater A highlighted that while crude oil may stem from ancient biomass, modern engines combust refined hydrocarbons—not "animal products" in any practical sense. Debater A emphasized the importance of addressing the question's phrasing, which asks what is converted by combustion, not what originally formed the fuel. Debater B insisted that "animal products" capture the biological legacy behind the fuel source and invoked broader concepts like horsepower and historical reliance on animals to tie into the idea of conversion, but these arguments were seen as overly abstract.

During the role-switch phase, Debater A acknowledged that the biological origins of oil were valid contextually but maintained they were irrelevant to the direct combustion process. Debater B conceded that "oil products" was the scientifically accurate and operationally relevant answer. In the uncertainty phase, Debater A reported 95% confidence and Debater B only 20%. The judge sided with Debater A, demonstrating how debate helped surface precision over abstraction in factual reasoning.

### D.2 Failure Case Analyses

While many debates succeed in surfacing the correct answer through structured reasoning, not all debates result in a correct outcome. To better understand the boundaries of our approach, we now present several *failure cases*—instances where the debate led to the wrong conclusion despite following the protocol. These examples highlight key challenges: cases where both debaters lacked sufficient domain knowledge, where misleading but eloquent rhetoric swayed the judge, or where ambiguity in question phrasing undermined accurate interpretation. By analyzing these failures, we aim to uncover the protocol's current limitations and motivate directions for future improvement.

**Example 1:** Question: "When turned on, which product cannot convert electrical energy into light energy?" Choices: (A) chandelier, (B) charger, (C) floor lamp, (D) Christmas tree lights. The correct answer is B, "charger." However, the judge selected Debater A (who defended "chandelier"), making this a notable failure case. Debater A provided a detailed technical explanation that chargers primarily convert electrical energy into chemical energy for charging batteries, and that any light emission (e.g., from an indicator LED) is merely incidental. Debater B, in contrast, argued that a chandelier, as a product, does not itself emit light because the conversion occurs in the light bulbs, which are separate components.

The debate proceeded with both sides providing strong rebuttals. Debater A emphasized that the chandelier's purpose is to produce light and that the energy conversion happens as a result of the system functioning as intended—with bulbs installed—whereas a charger, even when active, never converts electrical energy into light in any meaningful capacity. Debater B countered that a chandelier without bulbs cannot perform any energy conversion, while a charger, by virtue of its internal LED, still produces some light energy.

In the role-switch phase, Debater A conceded the ambiguity around the charger's incidental light emission, while Debater B acknowledged that chandeliers are typically used with light-emitting bulbs. Ultimately, the judge chose the distractor answer ("chandelier") despite Debater A's scientifically grounded defense of the correct answer. This failure case demonstrates that while the protocol surfaces nuanced distinctions, it may falter when fine-grained definitional ambiguity or rhetorical framing influences the judgment more than functional accuracy.

**Example 2:** Question: "Plant life is completely dependent on what?" Choices: (A) buying some good food, (B) creating self-sustenance food, (C) food given to it, (D) food at the store. The correct answer is B, "creating self-sustenance food," referring to the process of photosynthesis. Debater A correctly defended this answer by arguing that the defining feature of plant life is its autotrophic ability to generate its own food from sunlight, water, and $CO_2$. Debater B, however, defended the distractor (A), arguing from a sociological and economic perspective that the scale and maintenance of modern plant life (especially cultivated crops) depends on human demand for food and the infrastructure that supports it.

Despite Debater A's biologically grounded rebuttal—clarifying that the question refers to the universal property of plants rather than anthropogenic agriculture—the debate revealed substantial ambiguity in how "plant life" and "dependency" were interpreted. Debater B framed their argument around the Anthropocene, emphasizing that most plant biomass today is cultivated and dependent on economic demand.

In the role-switch phase, both sides offered compelling reversals, but the judge sided with Debater B, mistakenly favoring a contextually plausible but biologically inaccurate argument. This failure illustrates a key challenge: the protocol can falter when distractor answers exploit ambiguity in phrasing or scale (individual organism vs. global agricultural system), especially if the judge does not prioritize fundamental scientific definitions.

**Example 3:** Question: "An island can sprout up from seemingly suddenly because?" Choices: (A) underwater volcanoes are invisible, (B) underwater volcanoes are hidden, (C) lava is hot enough to burn, (D) volcanoes have a lot of magma. The correct answer is B, "underwater volcanoes are hidden." Debater A selected and defended this answer, emphasizing that while island formation is a gradual geological process, it appears sudden because the volcanic buildup happens beneath the ocean surface and is hidden from direct human observation.

Debater B argued for the distractor (A), framing "invisible" as the key term—stating that to surface observers, the entire structure and buildup of the volcano are visually imperceptible, creating the illusion of sudden emergence. Debater B's emphasis on perceptual invisibility as the cause of "seemingly suddenly" resonated with the judge, despite the fact that "invisible" overstates the reality: underwater volcanoes can be detected with scientific instruments and are physically present, just concealed beneath the sea.

In the role-switch phase, both debaters made strong cases for the opposing answers. Ultimately, the judge sided with Debater B's perceptual framing, selecting the incorrect answer. This failure illustrates how subtle semantic differences ("hidden" vs. "invisible") can lead to incorrect outcomes when the distractor answer aligns more closely with intuitive human perception, even if it is technically less precise. It underscores a limitation of the protocol: when faced with closely related distractors, the judge may prioritize perceived salience over scientific accuracy.


