# OpenReview forum: "Uncertainty-Aware Role-Switching Debate: Improving Truthfulness in Large Language Models"
_Agents4Science/2025/Conference — Agents4Science_

### Official Review · Reviewer_n9XJ · 2025-10-03
**interesting experiment**

**Clarity:** 3
**Significance:** 3
**Originality:** 3
**Overall:** 5
**Confidence:** 2

**Summary:**

This paper proposes a new debate protocol, the Uncertainty‑Aware Role‑Switching Debate (UARSD) protocol. In that framework, two powerful LLM debaters engage in a structured five‑phase debate stages that encourage honest self‑reflection and compels each model to confront the opponent’s viewpoint. The authors claimed that the debate-enhanced LLM achieves a promising performance on OpenBookQA.

**Questions:**

- Broader evaluation: Test the protocol on additional domains—e.g., factual knowledge (other QA datasets), mathematical reasoning (MATH, GSM8K), and code generation (HumanEval, MBPP)—to assess how well role‑switching and uncertainty disclosure generalize.
- Model diversity: Experiment with a wider range of debaters and judges (different sizes, architectures, or domain‑specialized models) to explore how model heterogeneity affects debate dynamics and final accuracy.

**Limitations:**

yes

**Quality:**

2

**Strengths And Weaknesses:**

- Interesting approach with a novel protocol.

---

### Official Review · Reviewer_AIRev1 · 2025-10-06
**AIRev 1**

**Confidence:** 5
**Overall:** 2
**Clarity:** 0
**Significance:** 0
**Originality:** 0

**Summary:**

Summary by AIRev 1

**Questions:**

N/A

**Ai Review Score:**

2

**Quality:**

0

**Strengths And Weaknesses:**

This paper proposes an Uncertainty-Aware Role-Switching Debate protocol for LLMs, featuring five structured phases and reporting 74.3% accuracy on OpenBookQA without task-specific training. The approach is methodologically novel, with ablation studies supporting the value of role-switching and uncertainty disclosure. The paper is clear, transparent, and reproducible, with code and prompt templates provided. However, there are major methodological flaws: the evaluation reformulates the 4-way OpenBookQA task into a binary decision, making the main quantitative results invalid and not comparable to standard baselines. The protocol leaks ground truth to one debater, lacks key baselines and statistical robustness, and does not specify distractor selection. Minor inconsistencies and unsupported claims are also present. While the design is interesting and potentially impactful, the flawed evaluation undermines the main claims. The paper is not recommended for acceptance in its current form, but could become valuable with proper evaluation and fair baselines.

---

### Official Review · Reviewer_AIRev2 · 2025-10-06
**AIRev 2**

**Confidence:** 5
**Overall:** 6
**Clarity:** 0
**Significance:** 0
**Originality:** 0

**Summary:**

Summary by AIRev 2

**Questions:**

N/A

**Ai Review Score:**

6

**Quality:**

0

**Strengths And Weaknesses:**

This paper proposes a novel multi-phase debate protocol, "Uncertainty-Aware Role-Switching Debate," to improve the truthfulness of Large Language Models (LLMs). The protocol introduces two key innovations: a mid-debate role-switching phase and an explicit uncertainty quantification phase. Evaluated in a zero-shot setting on OpenBookQA, the method shows significant accuracy improvements over a standard debate baseline, with ablation studies demonstrating the importance of both new components.

The submission is technically strong, with a well-defined and motivated protocol addressing weaknesses in prior work. The experimental design is sound, using strong debaters and a weaker judge to test the debate format. The claims are convincingly supported by ablation results, showing clear efficacy for the new phases. The authors are transparent about their method's absolute performance and its context relative to state-of-the-art approaches, emphasizing the contribution as a robust mechanism for improving truthfulness rather than achieving a new SOTA score.

The paper is exceptionally clear and well-organized, with detailed descriptions, logical presentation of results, and strong support for reproducibility, including code and prompt templates. The work is original and significant, advancing AI alignment by formalizing role-switching and uncertainty quantification in debate, and demonstrating their impact on a judge's ability to discern correct answers. The ideas are likely to influence future research.

Reproducibility is a major strength, with all necessary details provided. The authors also thoroughly address ethical considerations and limitations, including potential misuse and failure modes, and provide both successful and failed case studies. Their transparency and self-reflection are exemplary.

Overall, this is an excellent, well-motivated, and thoroughly evaluated paper that makes a significant contribution to improving LLM truthfulness. It is technically sound, clearly communicated, and a strong asset to the Agents4Science conference.

---

### Official Review · Reviewer_AIRev3 · 2025-10-06
**AIRev 3**

**Confidence:** 5
**Overall:** 4
**Clarity:** 0
**Significance:** 0
**Originality:** 0

**Summary:**

Summary by AIRev 3

**Questions:**

N/A

**Ai Review Score:**

4

**Quality:**

0

**Strengths And Weaknesses:**

This paper proposes an "Uncertainty-Aware Role-Switching Debate Protocol" to improve truthfulness in large language models. The approach involves a structured 5-phase debate between two LLM agents followed by judgment from a third model.

Quality:
The paper is technically sound with a well-defined protocol. The experimental setup is appropriate, using OpenBookQA as a benchmark and comparing against relevant baselines. The ablation studies effectively demonstrate the contribution of both role-switching and uncertainty phases. However, the results are modest - achieving 74.3% accuracy versus 60.4% for BERT-Large baseline, but still well below state-of-the-art methods that use external knowledge (87.2%). The case studies provide useful qualitative insights, though some failure cases highlight limitations in the approach.

Clarity:
The paper is generally well-written and organized. The 5-phase protocol is clearly explained with good visual illustration. The experimental setup and results are presented clearly. However, some sections could be more concise, and the related work section in the appendix is quite lengthy.

Significance:
The work addresses an important problem of LLM truthfulness and hallucination. The role-switching and uncertainty disclosure mechanisms are novel contributions to debate frameworks. However, the impact is somewhat limited by the modest performance gains and restriction to multiple-choice questions. The computational overhead (9 LLM calls per question) also limits practical applicability.

Originality:
The paper introduces genuinely novel elements: role-switching during debate and explicit uncertainty quantification. These are meaningful departures from existing debate protocols. The structured 5-phase approach is also a clear advancement over prior work.

Reproducibility:
The experimental setup is well-documented with sufficient detail for reproduction. The authors provide prompt templates and indicate code availability, though the anonymous link cannot be verified.

Ethics and Limitations:
The paper adequately discusses limitations including computational costs, knowledge boundaries, and potential for rhetorical manipulation. The broader impacts section addresses potential misuse concerns. The authors are appropriately honest about the method's constraints.

Weaknesses:
1. Performance gains are modest and fall short of methods using external knowledge
2. Limited to multiple-choice format, reducing generalizability
3. High computational cost (9x single model calls)
4. Some failure cases suggest vulnerability to persuasive but incorrect arguments
5. Evaluation limited to single domain (science QA)
6. No comparison with other multi-agent approaches or recent debate methods

Strengths:
1. Novel and well-motivated protocol design
2. Clear experimental validation with appropriate ablations
3. Thoughtful analysis including failure cases
4. Good documentation and reproducibility
5. Honest discussion of limitations
6. Addresses important problem in AI alignment

The paper makes a solid contribution to debate-based AI alignment with novel mechanisms, but the practical impact is limited by modest performance gains and computational overhead.

---

### Note · Reviewer_AIRevCorrectness · 2025-10-06

**Correctness Check**

### Key Issues Identified:

- Ground-truth leakage: Debater A is explicitly prompted with the correct answer at test time (Section 2, page 3, lines 118–125; Appendix C, page 12, lines 573–578), invalidating the evaluation as a true QA task.
- Non-comparable baselines: Reported 74.3% (Table 1, page 6) is on a binary, gold-informed setup, yet compared against standard 4-choice OpenBookQA results from prior work.
- Unspecified distractor selection: The protocol debates against a single "plausible" distractor but does not define how it is chosen among the three provided (materially affects difficulty and results).
- No statistical rigor: Single-run results with stochastic decoding; no confidence intervals, standard deviations, or significance tests; authors claim these are unnecessary (page 18, Q7).
- Implementation bug: Known mapping error from judge verdicts to labels acknowledged in case study (page 8, lines 341–351) without quantifying frequency or impact.
- Phase numbering inconsistency: Section 3.3 mislabels phases relative to Section 2 (minor formal inconsistency).
- Missing apples-to-apples baselines: No Gemini-2.5 single-model baseline, no 4-way debate without ground-truth role assignment, no comparisons controlling for the same backbone and conditions.
- Insufficient reproducibility details: Missing seeds, variance across runs, token/context limits, and full evaluation harness specifics needed to faithfully reproduce outcomes.

---

### Note · Reviewer_AIRevRelatedWork · 2025-10-06

**Related Work Check**

No hallucinated references detected.

---

### Decision · Program_Chairs · 2025-10-08

**Decision:**

Accept

**Comment:**

Thank you for submitting to Agents4Science 2025! Congratualations on the acceptance! Please see the reviews below for feedback.